# And Still She Rises: Policies for Improving Women’s Health for a More Equitable Post-Pandemic World

**DOI:** 10.3390/ijerph191610104

**Published:** 2022-08-16

**Authors:** Farah M. Shroff, Ricky Tsang, Norah Schwartz, Rania Alkhadragy, Kranti Vora

**Affiliations:** 1T.H. Chan School of Public Health, Department of Global Health and Population, Harvard University, Cambridge, MA 02138, USA; 2Maternal and Infant Health Canada, Vancouver, BC V6T 1Z3, Canada; 3Department of Medicine, Faculty of Medicine, University of British Columbia, Vancouver, BC V6T 1Z4, Canada; 4El Colegio de la Frontera Norte, Tijuana 22560, Mexico; 5Department of Medical Education, Faculty of Medicine, Suez Canal University, Ismailia 8366004, Egypt; 6Indian Institute of Public Health Gandhinagar, Gandhinagar 382042, India

**Keywords:** health policies, health equity, gender, pandemic, mental wellbeing, global health, COVID-19

## Abstract

The COVID-19 pandemic has spawned crises of violence, hunger and impoverishment. Maternal and Infant Health Canada (MIHCan) conducted this policy action study to explore how changes that have arisen during the COVID-19 pandemic may catalyze potential improvements in global women’s health toward the creation of a more equitable post-pandemic world. In this mixed methods study, 280 experts in women’s health responded to our survey and 65 subsequently participated in focus groups, including professionals from India, Egypt/Sudan, Canada and the United States/Mexico. From the results of this study, our recommendations include augmenting mental health through more open dialogue, valuing and compensating those working on the frontlines through living wages, paid sick leave and enhanced benefits and expanding digital technology that facilitates flexible work locations, thereby freeing time for improving the wellbeing of caregivers and families and offering telemedicine and telecounseling, which delivers greater access to care. We also recommend bridging the digital divide through the widespread provision of reliable and affordable internet services and digital literacy training. These policy recommendations for employers, governments and health authorities aim to improve mental and physical wellbeing and working conditions, while leveraging the potential of digital technology for healthcare provision for those who identify as women, knowing that others will benefit. MIHCan took action on the recommendation to improve mental health through open conversation by facilitating campaigns in all study regions. Despite the devastation of the pandemic on global women’s health, implementing these changes could yield improvements for years to come.

## 1. Introduction: COVID as Crisis and Catalyst

Worldwide, the COVID-19 pandemic has been an unprecedented shock. Many communities have not fared well during the crisis, largely due to socioeconomic inequities, which are magnified along gender lines. In some settings, increases in violence against women, food insecurity, isolation and mental health challenges have contributed to higher rates of all-cause morbidity and mortality (in this study, the term “women” defines all those who identify as women, encompassing transgendered, non-binary and gender non-conforming people). In many communities, women have borne the brunt of most of these challenges [1,2].

The COVID-19 pandemic has offered opportunities to improve global women’s health through analyzing the health disparities that apportion good health to those with more money and power and to provide possibilities for taking action to improve outcomes. One area in which this type of analysis is needed is women’s health before, during and after the COVID-19 pandemic. This dovetails with global feminist recovery plans that call for deep, transformative changes between the Global North and South and along gender lines to create a more socially just future with genuine solidarity across borders [3]. Hawaii’s plan calls for a shift from normative elitist masculine political culture to fostering systemic transformations toward a new economy that values women’s work and sustains families and communities. Supporting and incentivizing men to share caregiving responsibilities could reduce women’s fatigue [4].

Similarly, in Northern Ireland and Canada, various programs have called for interventions that decrease gender pay gaps, poverty, debt and the impacts of austerity on women’s health and reproductive care by instituting diverse voices in decision-making bodies and addressing racism and hate crimes [5,6].

In this context, Maternal and Infant Health Canada (MIHCan), which is a global public health collaborative that focuses on improving women’s health through research, education and innovation, conducted this study. We aimed to explore how changes that have arisen during the COVID-19 pandemic could catalyze improvements in women’s mental and physical wellbeing, with the goal of developing policy recommendations for a more equitable post-pandemic world. In this article, we analyze data from the COVID-19 Catalyst Study to identify areas to improve women’s health so that millions of people will not have suffered or died in vain. Many possibilities have emerged during this crisis and we explored three major areas that are related to mental and physical health and apply to our study regions.

To add an action component to this study, MIHCan also conducted a six-week campaign to foster open dialogues about mental health in each of the study regions. This emerged from our major finding that was related to the importance of improving mental health through open dialogue.

## 2. Methods

### 2.1. Study Design and Recruitment: Survey

Our research team was based in Canada, the USA, Mexico, India, and Egypt. The data for this policy research study were collected in a stepwise manner between September 2020 and March 2022. By utilizing a mixed methods approach, we collected qualitative and quantitative data in three phases. The first phase was an online survey, which was administered in English through Qualtrics. The survey solicited opinions on the role of women leaders during the COVID-19 pandemic and the potential successes for the wellbeing of women that were catalyzed by circumstances that occurred during the COVID-19 crisis. Team members recruited participants from their own regions using a combination of professional contacts and networks, via email, listservs, telephone and WhatsApp. A total of 280 participants completed the 15–20 min survey. The data from the surveys are reported elsewhere as this paper only reports on the focus group data.

### 2.2. Study Design and Recruitment: Focus Groups

A subset of the participants who completed the survey was subsequently invited to participate in focus groups. We chose focus groups because discussions amongst diverse researchers, clinicians and other women’s health experts could promote stimulating debate, dynamic engagement and immediate feedback and yield more insights than other methods. All focus groups were conducted online due to the ongoing pandemic and the dispersed geographical locations of the participants.

Collaboratively, we created the focus group questions and script (Appendix A). The questions were designed with the intent to generate policy ideas and recommendations. Co-investigators from different continents facilitated the selection of and follow-ups with the participants. By applying purposive sampling techniques, the focus group participants were selected for their expertise in women’s health in their geographical area. We recognized that most experts were from privileged backgrounds, with respect to socio-economic and educational factors, but we believed that their expertise in the domains of women’s health, public health and the status of women had the potential to yield valuable insights. The experts from the USA and Mexico focus groups were not as privileged and they had direct experience in farm workers’ organizations and related groups.

A pilot focus group was conducted in September 2020 with international public health professionals, which helped us to refine the focus group questions and script. We carried out another pilot focus group with our own research team after that, to further refine our questions. In total, a subset comprising 65 survey participants, including healthcare professionals, researchers and health policy experts, was recruited to participate in one of four online focus groups: Egypt and Sudan (*n* = 27); Canada (*n* = 18); the USA and Mexico (*n* = 12); India (*n* = 8). The demographic characteristics of the focus group participants are outlined in Table 1. All participants were required to complete or review the online survey prior to attending the focus groups as preparation and to stimulate talking points. The principal investigator (PI) moderated all focus groups while co-investigators were on hand to clarify language and cultural nuances, when required. All focus groups were conducted via videoconferencing. Zoom’s speech-to-text function was used to transcribe the dialog and all focus group sessions were video recorded with the consent of all participants. All data were password protected and stored on servers at the University of British Columbia in Vancouver, Canada.

### 2.3. Key Informant Interview

A series of key informant (KI) interviews were conducted with one Latinx woman in the USA to elucidate how the pandemic may have catalyzed better health in one of our study populations. To identify a KI from the working class Latinx community, we searched our networks. We looked for a woman who was willing to share her story in a video interview in either Spanish or English. We found a bilingual KI who was interviewed in English to maintain the consistency of the study language.

Claudia Lainez’s story offered context and insights from the perspective of a working class individual who has faced struggles for citizenship, human rights, safety from violence and good health. The KI interview validated the themes from the focus groups well. We felt that this one individual, who was interviewed twice for several hours, represented our themes exceptionally well.

#### 2.3.1. Data Analysis

The data were analyzed according to the framework method [7], from a feminist, anti-racism and decolonial lens. Research assistants reviewed the focus group transcripts for accuracy and any errors in the speech-to-text transcription were corrected by referencing the Zoom video and audio recordings. QSR NVivo 12 software was utilized for coding and thematic analysis by applying an iterative approach. The second author led the familiarization process and reviewed all of the corrected transcripts to generate a coding framework, which was corroborated by the other research assistants and investigators. The coding framework, once finalized, was transcribed into a standard codebook in NVivo 12. All transcripts and the standardized codebook were imported into NVivo 12 and two researchers coded each transcript manually and independently. Then, the overall κ coefficient was computed within NVivo 12 using the coding comparison query for each set of transcripts to assess intercoder reliability (ICR). A kappa coefficient threshold of κ ≥ 0.65 was established as the benchmark, which was then revised for any value below this until the predetermined threshold was met. The themes were articulated through an interrogation of the coded transcripts. This process was supplemented by comparing the transcripts using NVivo’s matrix coding query and visual comparison tool, which allowed for code charting and the subsequent identification of patterns. The codes were organized into major themes, which were informed by our research questions until consensus amongst the researchers was achieved to ensure the quality of the analytical process. Once the major themes were abstracted, they were subsequently reviewed for a final time by the researchers for congruency. During the coding process, direct quotes were identified by the researchers to illustrate the major themes that were identified in this study.

#### 2.3.2. Open Access to Data/Data Storage

All data will be available at https://maa.med.ubc.ca once this study has been fully published.

## 3. Results

We included four geographical areas in the study to represent global diversity and cross-border situations. Participants from nations in the Global North and South expressed similar ideas. This was not predicted as socioeconomic divisions between and within nations are significant. The four geographical areas in the study were chosen to represent global diversity: Egypt/Sudan; Canada; the USA/Mexico; India. Three major themes emerged across the study areas that suggested that the pandemic has created conditions for significant changes, which could lead to improving women’s health: (1) the need for open conversations and improved mental health status, (2) a greater need to support frontline workers; and (3) digital technology. Table 2 provides direct quotes that were associated with each of these themes.

Table 2 summarizes our results in words while Figure 1 does so with a mind map.

## 4. Discussion

This study addressed potential post-COVID-19 improvements in women’s mental and physical health and suggested policy recommendations for a more equitable post-pandemic world. Our team considered the seriousness of the global women’s health situation as the backdrop for this study. Every focus group included commentary about mental and physical health problems which have deteriorated during the pandemic. As a health policy action study, we strove for this to be a part of the solution to these problems. Our “Break the Stigma” campaign has shown that people in all of our study regions are eager to overcome shame and other barriers to improving mental wellbeing. We have learnt that this is a potent time for positive change.

The unmet needs of frontline workers, particularly those in the healthcare sector, emerged as a significant theme. Besides the need for childcare, PPE and hand sanitizer, the participants also discussed the need for the creation of breastfeeding stations in workplaces.

Adequate compensation for essential work through living wages, accompanied by benefits and protection, such as paid sick leave and the provision of PPE, are longstanding issues that have become more urgent due to the challenges of the pandemic. Worldwide, digital health technology has expanded at an unparalleled pace, creating flexibility for some women to work remotely or from home and opportunities to access clinical services via telemedicine.

Digital technology that supports working from flexible locations mainly benefits middle class women with adequate resources. Working class women who are frontline workers generally do not have the option to work from home. Beyond the digital divide, our participants critiqued and recognized the potential downfalls of digital health technology.

Most gendered research about the pandemic has revealed a disturbing picture of morbidity and mortality trends. In this increasingly polarized world, health inequities matter less to some leaders [8]. Deeply held patriarchal views are entrenched within governmental and institutional structures around the world. Research that has concluded that gendered health inequities are worsening is the first of many steps toward better health.

Globally, approximately 1 billion people experience mental health disorders, many of which are undiagnosed and untreated [9]. For many years, mental health has been relegated to the shadows of healthcare, partly due to stigmatization. This extensive suffering poses human costs to patients, informal caregivers (most of whom are women) and society at large. Worldwide, due to anxiety, depression and other mental health conditions, approximately USD 2.5 trillion a year is lost to reduced productivity, absenteeism and ongoing morbidity [10]. Our participants believed that a greater emphasis on mental health and wellbeing, additional support for frontline workers and expanded digital technology could improve the overall wellbeing and mental health status of women.

Globally, mental health status has declined during the pandemic, partially as a result of isolation, which has led to loneliness [11]. Loneliness is correlated to all-cause mortality, including increased vulnerability to coronary heart disease, depression and other diseases that contribute to lower quality of life [12,13,14]. Anxiety and depression have increased substantially during the pandemic [15] as many people were forced into isolation, even for significant events in their lives, such as birthdays, funerals and weddings. Worldwide, this unnatural social order has led to all-age rises in mental health concerns, particularly among women [15], who have been burdened with extra caregiving, cooking, cleaning and emotional labor, often at the expense of their paid work [16].

Living wages, paid sick leave and the provision of PPE were prioritized by our participants as support for frontline workers. These provisions could address the social determinants of (women’s) health and have been found to reduce absenteeism, improve retention, increase productivity, lower insurance costs and, ultimately, lead to a better quality of life [17].

### 4.1. Telemedicine and Telecounseling

Telecounseling has burgeoned during the pandemic by providing a long-awaited service for patients who are unable to or are greatly inconvenienced by leaving their homes [18]. The participants from each of our focus group regions concurred that telemedicine improved women’s access to care, as long as interpretation services were provided when necessary. Various kinds of telemedicine have been established around the world that permit those without digital devices to access clinicians at a distance.

Our participants felt that telecounseling could provide a possible solution to mental health problems, currently and moving forward. Organizations were urged to emerge from this pandemic with an improved capacity to compassionately hear mental health concerns. Our participants overwhelmingly agreed that we ultimately need a new world that is built on respect, love and human interconnection. When discussing the silver linings that were related to mental health issues, the participants noted an increase in patience, kindness and support during these challenging times of isolation when lockdowns exacerbated violence and existing mental health conditions.

This crisis has also prompted creative solutions. In India, for example, where many working class people operate in the informal sector and sell goods and services, it was found that: “The pandemic provided an opportunity for people to take a little more of primary healthcare into their own hands. More self-care, a bit of guidance and people then took over, especially with things like contraception and pregnancy testing.” Our Indian participants told us that women’s work in the home increased manyfold in households where hired staff usually provided support. The benefits of this otherwise taxing burden were that male family members observed these wearisome efforts to manage a household for the first time and, therefore, contributed. This could potentially be the beginning of a much-needed shift in the burden of household labor. For decades, many have expressed the need for task-sharing in the domestic sphere [19] and the pandemic has made some of this possible.

The participants in big cities with congested traffic, such as Delhi, Los Angeles, Khartoum and Cairo, commented on saving two to three hours daily and using this time for family, self-care and a better work–life balance. Many participants believed that flexible work locations ought to be permanently instituted. Emotional contact with others was also facilitated through technology. Working from home created some extra burdens, but these were outnumbered by the benefits, according to our participants.

Telemedicine has flourished during this time and has prompted innovations in medical practice. It is cost-effective and more convenient, particularly for those who experience difficulties in accessing services. Both clinicians and patients benefit from telemedicine and higher satisfaction ratings have been reported for video-enabled consultations [20]. Telemedicine has also reduced the possibilities of COVID-19 exposure by limiting patient–provider contact [21]. Consistent with the evidence about online mental health counseling, our participants noted that telemedicine is effective, particularly because those with serious conditions could be incapacitated and unable to access in-person care.

#### Concerns Related to Digital Technology

While our participants were generally in support of expanding access to digital technology to meet health needs, such as telemedicine, telecounseling and flexible work locations, we believed that a critical lens on technology was also in order. Technological solutions present various concerns, including the digital technology gap, the inability to replace human contact, the replacement of human workers by artificial intelligence, musculoskeletal and other health problems, as well as concerns about human rights related to surveillance. We address some of these concerns below.

Filling the digital technology gap is a critical concern. All over the world, some people are unable to access basic information because they lack digital devices. Some governments, such as those in Canada, plan to bridge the digital divide for people in rural areas and those with limited incomes. Such initiatives could improve access to information for all people, including people with disabilities, the elderly and those whose income or geography limits their digital access [22]. Training to use digital technology must also be part of these packages, especially training for women.

In cases where smartphones may not be feasible, it is still possible to provide access to information and services using telephones or other simple technologies. Many clinic visits during the pandemic occurred over telephones and not through video-enabled platforms [23]. Care at a distance for all visits is generally not optimal and physicians’ satisfaction rates have been lower than those of patients because they are aware of the limitations that it places on their ability to create positive relationships and to diagnose and treat conditions [24].

In our study regions, 93% of Egyptians have mobile phones [25], while 76% of Sudanese adults have mobile phones [26] and 98% of Canadians have mobile phones [27]. In the USA, 85% of Latinx adults have smartphones, which matches the national average [28]. Currently, 79% of Indians have access to smartphones, although this percentage promises to increase in the future [29].

In our study regions, mobile phones are thus viable options for improving women’s health, as long as women are able to access the devices, which are often owned by male household heads. Given the above figures, between 5–24% of people may not be able to access services, but these percentages may be higher for women.

As well as the digital divide, other challenges that are related to digital technology include mental and physical side effects. Screen dependency disorder has been a growing concern for many years but during the pandemic, many people’s lives were largely mediated through screen interactions. As it disproportionately impacts children and youth, screen dependency disorder has been diagnosed in 2–12% of young people worldwide [30].

Moreover, musculoskeletal problems are also increasing because of excess hours in front of screens, as well as poor ergonomics that inevitably occur in home offices [31]. Neck, shoulder, upper and lower back, hand and wrist problems, among others, have increased during the pandemic, which have led to other health problems and a loss of productivity. Eye health concerns that are related to screen use are often ignored, such as dryness, fatigue, irritation and loss of focus, which can lead to sleep problems and other health issues [32].

Wireless communication technologies have also been linked to many disorders, including cardiovascular disorders, neurodegeneration, metabolic and sleep disturbances, cancers and other conditions [33]. Moreover, technology, even virtual reality, does not adequately replace in-person care. Providing virtual care 100% of the time is unlikely to be equivalent to in-person care, as noted by one of our participants:

“There’s some compromises being made in telemedicine and in-person care is still better.”(USA/Mexico)

Furthermore, misdiagnoses may be more likely to occur via telemedicine, which could lead to a cascade of medical errors, such as incorrect medication consumption and other serious problems [34]. In-person follow-ups are crucial after telemedicine consultations. Additionally, some physicians believe that telemedicine could negatively affect patient relationships and decision-making and increase the burden of care [35]. Data privacy issues have only grown larger as telemedicine has burgeoned.

Finally, technological dystopias with excessive surveillance have already arrived in some parts of the world. Indeed, 5G networks and other supercharged technologies have made it possible to spy on people intensely, quietly abusing our human right to privacy. Encouraging more technology inevitably promulgates this dangerous trend toward a technocratically governed society, especially when attempts to regulate it are absent [36]. Finally, surveillance technologies have been linked to violence against women through stalking, data tracking and more [37].

## 5. Health Policy Recommendations

Our recommendations emerged from the ideas that were the most pressing and frequently mentioned ideas among our participants, combined with our contributions and evidence from scholarly literature. Figure 2 represents these ideas in schematic form. As hoped, the focus groups generated a rich stream of dialogue and provided insights into the needs of women through the lens of expert panels on womens’ health. This was augmented by a key informant interview with a working class Latinx woman who discussed themes, challenges and needs that corroborated with our focus groups. The three major findings we elucidated were the **importance of promoting mental wellbeing through open conversations, the necessity to value frontline workers through living wages, expanding employee benefits and providing PPE to protect their wellbeing and the need to leverage digital technology to expand remote working options and access to healthcare through telemedicine and telecounseling.**

These recommendations could benefit many people, with a disproportionate impact on those who identify as women. Mental health burdens, for example, are greater for those who identify as women [38]. Similarly, most frontline workers in healthcare, retail and hospitality are women, who have been harshly impacted by the economic consequences of the COVID-19 pandemic [39]. Digital technology could likewise benefit white-collar women workers differently than others because most domestic and childcare labor falls on women. Finally, telemedicine and telecounseling could also significantly benefit women because there is less of a need to arrange for others to care for children and loved ones when women can receive services from home.

### 5.1. For Workplaces and Communities

Workplaces and communities should promote mental wellbeing through initiatives that encourage safe and open dialog in an attempt to end stigma. Our recommendations include:Create wellbeing initiatives to foster cultures of safety in which people can feel comfortable and not vulnerable when discussing mental health concerns. When leaders openly identify their own struggles with mental health, it can create safety for others to do so as well. Safeguarding job stability and retention within this open environment would be critical for improving workplace conditions for those with mental conditions. Workplace wellness programs, stress reduction initiatives and mental health courses have proven to reduce absenteeism and improve retention and productivity [40].Enlist the voices of social icons who are willing to disclose their mental health concerns, which could assist in changing social norms. Film stars, athletes, intellectuals and other social influencers ought to be encouraged to lead by example and openly disclose their mental health stories. When people who hold status in society freely discuss their difficulties with mental health issues, others can be inspired to accept the realities that are posed by mental health anguish.Midwives and other birthing professionals ought to highlight the importance of mental wellbeing for their clients as mental health issues from 0–6 years are highly impactful on population wellbeing [41]. Taking care of the mental health needs of pregnant women, such as depression screening [42], and encouraging them to begin the lives of the next generation with openness around mental wellbeing through parenting that attends to mental wellbeing could make positive intergenerational impacts. Policies in hospitals, clinics, professional associations and educational institutions that require birthing professionals to be well-trained in evidence-based mental health counseling ought to be instituted.Through policies in training institutions and workplaces, early childhood teachers, care providers and guardians ought to be encouraged to implement learning modules for young children that are aimed at developing mental health literacy and the normalization of open conversations around mental health status. This openness could encourage the expression of all feelings, including joy, peacefulness, sadness, anger and others.

Ultimately, these efforts could promote micro- and macro-culture shifts that place mental health conditions on par with (non-stigmatized) physical health conditions.

### 5.2. For Employers

Employers should value frontline workers through improvements to workplace environments, including living wages and employee benefits, as well as the implementation of worker-centered and dynamic work arrangements that are reflective of modern society. Our recommendations include:Provide living wages and worker-centered benefits (including affordable childcare and paid sick leave) and ensure the adequacy and sufficiency of PPE and other safety equipment for frontline workers. The equitable distribution of PPE has become a basic right for frontline workers for the remainder of this pandemic. When it is over, PPE will still be important for protecting frontline workers against other airborne conditions.Where possible, provide flexible and inclusive workplace options, such as working from home, adaptable schedules, hybrid models, onsite childcare and lactation rooms.

### 5.3. For Governments and IT Providers

Governments and IT providers should develop the role of digital communication and address the socioeconomic and health disparities by bridging the digital divide. Our recommendations include:Invest in modern infrastructures for the widespread provision of reliable, accessible and affordable internet services and devices.Fund digital literacy training and technical support for new users of digital communication and technology.

### 5.4. For Health Service Organizations

Where clinically indicated and appropriate, health service organizations should provide telemedicine options for physical and mental health conditions to broaden the reach of healthcare services. Our recommendations include:Invest in patient-friendly digital infrastructures that are accessible on a variety of devices.Engage and educate patients about digital alternatives to in-person care.

## 6. Health Policy Implementation Efforts

In the interests of knowledge translation and the implementation of policy in action, team members approached policymakers to discuss these recommendations, including talks with technology companies to provide free devices and subsidized data and talk-time for women with low incomes. The authors were keen to realize some concrete changes as a result of the study.

In Egypt, one of the authors conducted sessions to help university staff to cope with stress and ensure their wellbeing post-pandemic through flexible work schedules. She is also making efforts to develop accessible daycare in hospitals for frontline workers.

In all study regions, Maternal and Infant Health Canada (MIHCan), which facilitated this study, has taken initiatives to create open dialog about mental health and wellbeing. MIHCan introduced the “Break the Stigma” program, which is a mental health campaign on social media channels, with the goal of destigmatizing conversations around mental health.

We also proposed a mental wellbeing app called *Ananda*, which aims to support the mental health needs of Indian adolescents. *Ananda* will be free and anonymous and uses games, a discussion board and traditional Indian wellness methods, such as yoga, meditation and Ayurveda.

## 7. Study Limitations

The limitations of consulting a pool of experts who were located in relatively limited geographical sites were related to possible subject position biases. Most of our participants identified as women and worked as clinicians, educators and researchers, among others. Our Mexico/USA focus group was an exception, as most of the participants were not as privileged as the experts in other focus groups. Yet Canadian participants mentioned their privileged subject positions:

“Acknowledging my privilege that I’m able to work from home… I’m not a frontline worker. I’m not a low wage worker in a high risk …environment.”(Canada)

“I don’t think this is unique to women, but I think it’s very much about disability and, of course, many women are disabled. I also am privileged enough that I can work remotely with my job…That’s actually meant that for the first time in my career, my job is actually accessible for me.”(Canada)

Regardless, as much of their work has been with patients, clients and research respondents who are vulnerable, our recommendations reflected the importance of addressing working conditions for frontline workers. Our key informant interviews (available at maa.med.ubc.ca) with a working class Latinx woman confirmed virtually all of the points that were raised in the focus groups, which were related to the importance of mental wellbeing, technology, time for self-care and working from home.

Conducting the focus groups in English was another limitation, particularly for some of our participants in Egypt and Sudan. Although a co-investigator from the region who was fluent in Arabic was present to clarify language and cultural nuances where appropriate, the possibility for incomplete transmissions of intent and experiences remained.

## 8. Conclusions

Most studies on gender and COVID-19 have examined the tragic decline of women’s mental and physical health status. In contrast, our study analyzed how the pandemic catalyzed improvements in women’s health across disparate regions of the world and demonstrated significant similarities. We conducted a global survey and focus groups of women’s health experts in Egypt and Sudan, India, Canada and the USA and Mexico, thereby recognizing the importance of women’s health for the world:

“When we uplift women, women uplift the world.”(Canada)

As hoped, the focus groups generated a rich stream of dialogue and provided insights into the needs of women through the lens of expert panels on women’s’ health. This was augmented by a key informant interview with a working class Latinx woman, who discussed themes, challenges and needs that corroborated with those that were identified by our focus groups. **The three major findings that we elucidated were the importance of promoting mental wellbeing, the need to leverage digital technology to expand the reach of healthcare and remote working options and the necessity to value frontline workers by expanding employee benefits and providing PPE to protect their wellbeing.**

Stopping everyday routines during lockdowns offered people the opportunity for profound reflection about their purpose and meaning, initiating significant life changes, such as changing jobs or locations. Yet, this game-changing pandemic has exacerbated mental health problems for many people, prompting us to recommend widespread open disclosure about mental health concerns as a trend to lift stigma. This is not a novel idea; it is already in place in various institutions, but it needs expansion. Our participants envisioned societies that supported mental and social wellness. MIHCan implemented the “*Break the Stigma*” (https://maa.med.ubc.ca/break-the-stigma-campaign/) program, which is a campaign to encourage mental health conversations, and is encouraging other organizations to follow suit.

This era has catalyzed the reinvigoration of community support for tasks such as food provision, medication refills, etc. Claudia Lainez (our key informant) and her daughter contracted COVID-19 prior to the advent of the vaccines. Their experience with the virus was unlike dominant media reports. They received a tremendous amount of support from family and friends, so they ate well and visited people outdoors. All of their needs were met. Fortunately, they fully recovered, partly due to the caring and kindness around them.

Resonant with feminist recovery plans, we recommend the revaluation of women’s labor. Among our recommendations is the need for optimum compensation for frontline workers with benefits, such as childcare, paid sick leave and PPE. Moreover, we recommend the large-scale permanent provision of options for workers to have flexible work locations and, where clinically appropriate, provide telemedicine and telecounseling. The growing use of technology has produced a digital divide that disproportionately impacts racialized women who live in poverty. Providing access to the web, devices and training would assist in bridging the digital divide. Problems related to mental and physical wellbeing pose significant threats to human rights.

Despite these actions, we recognize that policies can only go so far. Unequal structures need to be replaced with equitable gender-just systems, as expressed by our Canadian participants:

“We need a world that’s based on equity and equality for… all people [who] identify as women and we need something where there are basic incomes, where discrimination and systemic racism and the factors that marginalize women along all of the intersectional lines are mitigated and nonexistent.”

“Addressing systemic racism has to happen at all levels: municipal, provincial, national and global.”

“Transforming the system will allow us to get at what are the policies that need to be in place to be able to maintain that transformation.”

We plan to conduct a follow-up study on transformation and women’s health that will mainly focus on the expertise of working class women.

While the COVID-19 pandemic has presented the world with calamity and has taken women’s health and human rights back some decades [43], we optimistically believe that this crisis *can* catalyze positive changes in women’s health. Given the enormity of our health needs, fostering systemic changes that can improve the wellbeing of half of the global population will yield many positive returns. Retreating to the former status quo would be a poor option for women. The time is now for governments, businesses and civil society organizations to recreate our world for better women’s health.

## Figures and Tables

**Figure 1 ijerph-19-10104-f001:**
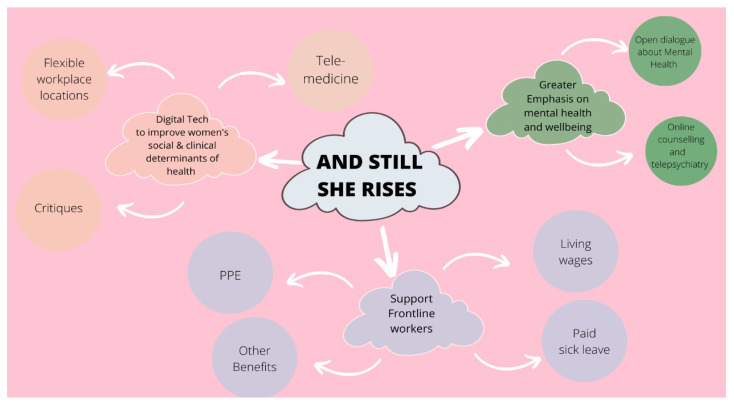
The focus group themes across geographic regions (Diagram by Sheyn Hosanee).

**Figure 2 ijerph-19-10104-f002:**
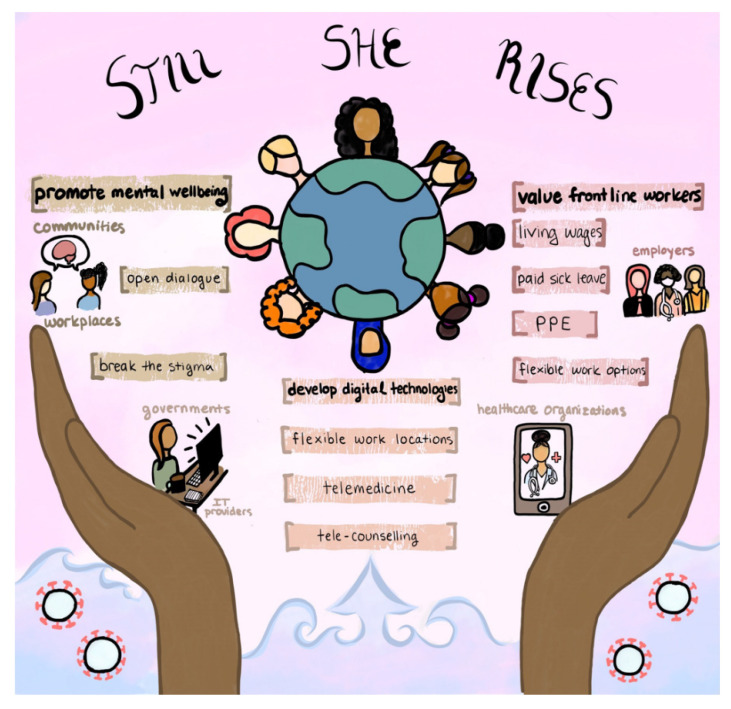
Our women’s global health policy recommendations (Diagram by Jaya Kailley).

**Table 1 ijerph-19-10104-t001:** The demographic characteristics of the focus group participants.

Demographics	Canada	USA/Mexico	India	Egypt/Sudan
Sex, *n* (%)				
Female	18 (100)	12 (100)	5 (62.5)	21 (78)
Male	0 (0)	0 (0)	3 (37.5)	6 (22)
Other	0 (0)	0 (0)	0 (0)	0 (0)
Primary Profession, *n* (%)				
Clinician/Physician	3 (17)	0 (0)	2 (25)	3 (11)
Researcher	6 (33)	3 (25)	2 (25)	8 (30)
Educator	4 (22)	3 (25)	0 (0)	7 (26)
Public Health	1 (5.5)	2 (17)	4 (50)	4 (15)
Student	1 (5.5)	1 (8)	0 (0)	3 (11)
Other	3 (17)	3 (25)	0 (0)	2 (7)

**Table 2 ijerph-19-10104-t002:** A summary of the themes with corresponding direct quotations from the focus group participants.

Theme	Direct Quotation
1. Open Conversations about Mental Health	“When a mother bonds with their child, you think about their wellbeing so as a parent I think about my children’s wellbeing. But then as a colleague I think of my colleagues’ wellbeing and also my friends and family throughout this pandemic, not only their physical, but their mental health and resiliency to be able to handle those things. And I feel like the voices of women have been very empowering in this time, where we find so much in the USA and other countries such as racial tension.” (USA/Mexico) “[Since the pandemic] I’ve seen patience increase over a lot of different conversations. Also, a lot more conversations about empowering yourself to be your own advocate, whether it be your physical or mental health.” (USA/Mexico) “A lot of psychological problems are not even considered; they have been overlooked. If a woman is breaking down psychologically it’s not something to consider.” (Egypt/Sudan) “My job is to help people’s light shine. Each of us was born with a light.” (Canada) “That’s really what this is all about, [it’s] about love. If we could just love each other and treat each other with respect and dignity, what a better world we’d be in.” (Canada) “Speaking from a very New York City perspective I have to say that the most empowering people that I’ve worked with during the pandemic have been …women, working as leaders in community based organizations, faith based organizations, thinking about the wellbeing of people, almost like how mothers think about their young and their wellbeing and how that instinctual nature that comes into play when a mother bonds with their child.” (USA/Mexico)
2. Supporting Frontline Workers	“My…colleagues are working frontline…hospitals, quarantine departments and everything. I am really heartbroken to see them struggling. They are really concerned about their children. Some of them have really young children, still they don’t have the luxury of taking leave.” (Egypt/Sudan) “85% of frontline workers are women, who… are not given enough PPE, not been given enough sanitizers. So that is a big gap. Most of the PPE is at the hospital level with senior doctors.” (India) “I was pleasantly surprised to realize that in the early days of the pandemic… frontline health staff did not panic. They just held their ground… dug in their heels and stuck it out. Even though these were bad times. That was positive–the women as the men.” (India) “How to protect frontline workers and the very particular needs that women have…in terms of women’s motherhood needs–breastfeeding. Our people wanted to continue breastfeeding [while working].” (Egypt/Sudan) “In our setting, we created breastfeeding pods that were very attractive and were used a lot. Mothers were happy to feed their babies while at the hospital.” (USA/Mexico) “Women always have to bear the burden of feeding their children and keeping the house. They took more risks than others, to see that as soon as work was available, they started doing work. And it could be in any sector. I mean, from vegetable selling to [anything else]. The sectors in which women were working, …the work started earlier. In terms of rebooting, women took the lead.” (India) “You’re talking about pay equity and…employment equity, hiring more women and then paying them [well]. Yeah, equal wages. Mm hmm.” (USA/Mexico) “Women have been severely disadvantaged during this pandemic, particularly women of color, low income women and people who are in frontline jobs. This also presents us with an opportunity to do that type of disruption right to start —tearing down some of these structures that have built up around us for so long. Never let a good crisis go to waste.” (Canada) “The pandemic provided an opportunity for people to take a little more of primary healthcare into their own hands. More self-care, a bit of guidance and people then took over, especially with things like contraception and pregnancy testing.” (India) “[In] many places, PPE was lacking, even in the bigger hospitals. They supplied everyone with raincoats and they taped it from all the possible points, where the holes are. So, it’s…a very good innovator, you know, local approaches that have happened and I think there will be 1000 other examples, where even though things were not available, people have…thought about what can be done, in [sub]optimal conditions.” (India)
3. Digital Technology	“Ensure that people who normally wouldn’t have technology, including people with disabilities [and] seniors, have the technology [and] develop the skill set to be able to access programming online” (Canada) “You have some time to enjoy your habits; you enjoy your family; enjoy your life. I think it was a very rich experience especially for me.” (Egypt/Sudan) “I think it’s going to be very hard to go back to the, no virtual options and everything in person.” (Canada) “I really enjoyed working from home. I felt that I had the chance to take care of my kids and work at the same time.” (Egypt/Sudan) “One of the concerns…I have is that all of this extra access, all this additional accessibility…I’m worried it’s going to disappear.” (Canada) “There has been greater wellbeing for women during this pandemic since we are working from home.” (Egypt/Sudan) “Three hours now open in the day that we are able to fill either with self-care practices or spending more time with our families but also getting more involved in our community— organizing and so on.” (USA/Mexico) “[Things would be better] if there were reliable transportation systems for people that work.” (Mexico/US) “I [am] so pleased that I can…spend time with my kids at the…same time working from home and…be[ing] productive.” (Egypt/Sudan) “This gives us more time to enjoy being with our children, enjoying … quality time with them…I think in the future, … this can be applied.” (Egypt/Sudan) “Optimizing technology has been a silver lining. I think we have all developed new skills because of that. [Before], it was mostly in-person training. Now we have…access to international training… [and] information at the tip of our hands, …on websites.” (USA/Mexico) “Why not save time? All this effort… just to reach your office.” (Egypt/Sudan) “I think the need of contacting others, especially women to women: mothers, sisters, cousins, friends, [the] kind of push that [video] technology offers so we can communicate, see each other, laugh together. It’s beyond just the phone call.” (USA/Mexico)

## Data Availability

All data will be available at: https://maa.med.ubc.ca once the study has been fully published.

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
