# Peer review of "And Still She Rises: Policies for Improving Women’s Health for a More Equitable Post-Pandemic World"

_ijerph, 2022, doi:10.3390/ijerph191610104_

Round 1

Reviewer 1 Report

Dear Authors.

Thank you for your valuable manuscript.

Please do the following improvements:

1. Please correct the first affiliation (missing "r" in the Harvard word).

2. Please adapt your abstract to the journal's standard format: "The abstract should be a total of about 200 words maximum. The abstract should be a single paragraph and should follow the style of structured abstracts, but without headings: 1) Background: Place the question addressed in a broad context and highlight the purpose of the study; 2) Methods: Describe briefly the main methods or treatments applied. Include any relevant preregistration numbers, and species and strains of any animals used. 3) Results: Summarize the article's main findings; and 4) Conclusion: Indicate the main conclusions or interpretations. The abstract should be an objective representation of the article: it must not contain results which are not presented and substantiated in the main text and should not exaggerate the main conclusions."

4. Please expand abbreviations when used for the first time in the text (also applies to the abstract section).

5. I do not understand the portrayal/comparison of the impact of x-rays for diseases diagnosis to COVID-19 disease (lines 45-48).

6. Please provide the citations according to the Journal style: "In the text, reference numbers should be placed in square brackets [ ], and placed before the punctuation; for example [1], [1–3] or [1,3]."

7. Please improve the references according to the Journal style:

"The reference list should include the full title, as recommended by the ACS style guide. Style files for Endnote and Zotero are available. References should be described as follows, depending on the type of work:

  • Journal Articles:
    1. Author 1, A.B.; Author 2, C.D. Title of the article. Abbreviated Journal Name Year, Volume, page range.
  • Books and Book Chapters:
    2. Author 1, A.; Author 2, B. Book Title, 3rd ed.; Publisher: Publisher Location, Country, Year; pp. 154–196.
    3. Author 1, A.; Author 2, B. Title of the chapter. In Book Title, 2nd ed.; Editor 1, A., Editor 2, B., Eds.; Publisher: Publisher Location, Country, Year; Volume 3, pp. 154–196.
  • Unpublished materials intended for publication:
    4. Author 1, A.B.; Author 2, C. Title of Unpublished Work (optional). Correspondence Affiliation, City, State, Country. year, status (manuscript in preparation; to be submitted).
    5. Author 1, A.B.; Author 2, C. Title of Unpublished Work. Abbreviated Journal Name year, phrase indicating stage of publication (submitted; accepted; in press).
  • Unpublished materials not intended for publication:
    6. Author 1, A.B. (Affiliation, City, State, Country); Author 2, C. (Affiliation, City, State, Country). Phase describing the material, year. (phase: Personal communication; Private communication; Unpublished work; etc.)
  • Conference Proceedings:
    7. Author 1, A.B.; Author 2, C.D.; Author 3, E.F. Title of Presentation. In Title of the Collected Work (if available), Proceedings of the Name of the Conference, Location of Conference, Country, Date of Conference; Editor 1, Editor 2, Eds. (if available); Publisher: City, Country, Year (if available); Abstract Number (optional), Pagination (optional).
  • Thesis:
    8. Author 1, A.B. Title of Thesis. Level of Thesis, Degree-Granting University, Location of University, Date of Completion.
  • Websites:
    9. Title of Site. Available online: URL (accessed on Day Month Year).
    Unlike published works, websites may change over time or disappear, so we encourage you create an archive of the cited website using a service such as WebCite. Archived websites should be cited using the link provided as follows:
    10. Title of Site. URL (archived on Day Month Year)."

10. Please split the results and discussion section into two separate paragraphs, in this form the results are unreadable.

11. Please consider adding the following publication to the literature: https://doi.org/10.3390/ijerph19010180

Author Response

Reviewer 1 Comments

Author Response

1. Please correct the first affiliation (missing "r" in the Harvard word).

Harvard is now spelled correctly, in the author affiliation section.

2. Please adapt your abstract to the journal's standard format: "The abstract should be a total of about 200 words maximum. The abstract should be a single paragraph and should follow the style of structured abstracts, but without headings: 1) Background: Place the question addressed in a broad context and highlight the purpose of the study; 2) Methods: Describe briefly the main methods or treatments applied. Include any relevant preregistration numbers, and species and strains of any animals used. 3) Results: Summarize the article's main findings; and 4) Conclusion: Indicate the main conclusions or interpretations. The abstract should be an objective representation of the article: it must not contain results which are not presented and substantiated in the main text and should not exaggerate the main conclusions."

We have adapted the abstract using the headings and reduced it to about 200 words. It is now a single paragraph.

Please expand abbreviations when used for the first time in the text (also applies to the abstract section).

In the abstract we have written “Maternal Infant Health Canada (MIHCan)”

We have also written “Personal Protective Equipment (PPE)”

We have ensured that all abbreviations are expanded when used for the first time.

I do not understand the portrayal/comparison of the impact of x-rays for diseases diagnosis to COVID-19 disease (lines 45-48).

We have deleted this sentence fragment “Yet, just as an X-ray helps identify lesions and areas for potential intervention”

Please provide the citations according to the Journal style: "In the text, reference numbers should be placed in square brackets [ ], and placed before the punctuation; for example [1], [1–3] or [1,3]."

All the citations are fixed and they now are according to journal style.

 Please improve the references according to the Journal style:

"The reference list should include the full title, as recommended by the ACS style guide. Style files for Endnote and Zotero are available. References should be described as follows, depending on the type of work:

The references are now according to the journal style.

Please consider adding the following publication to the literature: https://doi.org/10.3390/ijerph19010180

Thank you for informing us about this article related to OB/GYN services in Poland during the pandemic. We have cited the article and added it to the references list. The article mentions that women’s reproductive health services were challenged during COVID-19 and offers recommendations on how to meet the challenges. In reading the article carefully, we saw that the conclusion in the Polish paper regarding depression screening is for midwives to screen pregnant women was relevant to one of our recommendations so we have added the citation there.

You can find it in Health Policy Recommendations, #3: “Taking care of the mental health needs of pregnant women, such as depression screening (Wszolek et al, 2022) and encouraging them to begin the next generations’ lives with openness related to mental wellbeing through positive parenting will make intergenerational positive impacts. Policies in hospitals, clinics, professional associations, educational institutions that require birthing professionals to be well-trained in evidence-based mental health counseling ought to be instituted.”

Please split the results and discussion section into two separate paragraphs, in this form the results are unreadable.

We have now split the results and discussion into two sections with clear delineation between the two. Further, we have collated our supporting quotations into a single Table as part of our results.

Reviewer 2 Report

Review the manuscript entitled “And Still She Rises: policies for improving women’s health for a more equitable post-pandemic World” for IJERPH.

The present study has evaluated two hundred and eighty experts in women’s health, and 65 subsequently participated in focus groups from India, Egypt, Sudan, Canada, the United States, and Mexico. The objective was to conduct a policy action study to reimagine better health, despite setbacks worldwide for women. The key findings included the need to value and compensate those working on frontlines; digital technologies have created flexibility to work from home; Telemedicine and tele-counseling also offer greater access to care. Based on these findings, the authors recommend some policies that apply to the study regions. It is an important manuscript for improving the health disparities that must be an international priority.

Minor concerns:

1.      Introduction: I think something is missing in the last sentence, so I suggest rewriting it.

To add an action component to this study, MIHCan established a six-week campaign to foster open dialogs about mental health in each of the study regions. In this manuscript, we analyze data from the COVID-19 Catalyst Study to identify areas for promising shifts to improve women’s health --so that millions of people will not have died in vain.”

2.      Methods:

a.      It is important to include the social demographic characteristics of the participants such as age, gender, and occupation.

3.      Conclusion:

I think the authors could summarize in a few sentences the main findings, as we can observe in figure 2.

Author Response

Reviewer 2 Comments

Author Response

Introduction: I think something is missing in the last sentence, so I suggest rewriting it.

“To add an action component to this study, MIHCan established a six-week campaign to foster open dialogs about mental health in each of the study regions. In this manuscript, we analyze data from the COVID-19 Catalyst Study to identify areas for promising shifts to improve women’s health --so that millions of people will not have died in vain.”

We added in this sentence “Many silver linings emerged during this crisis and we explore three major areas related to mental and physical health that apply to our study regions.”

Replaced the last sentence in the introduction with “To add an action component to this study, MIHCan also conducted a six-week campaign to foster open dialogues about mental health in each of the study regions. This emerged from our major finding related to the importance of improving mental health through open dialogue.” In brief, the introduction has been reconsidered and revised.

It is important to include the social demographic characteristics of the participants such as age, gender, and occupation.

We appreciate this suggestion. We did recruit  a range of people with diverse demographic characteristics.

 Demographic characteristics of focus group participants are documented in Table 1, a newly created table.

I think the authors could summarize in a few sentences the main findings, as we can observe in figure 2.

Thank you for this suggestion. We have included a summary in the health policy recommendation section.

As we had hoped, the focus groups generated a rich stream of dialogue and provided insight into the needs of women through the lens of expert panels on womens’ health. This was augmented by a key informant interview of a working class Latinx woman who discussed themes - challenges and needs - that corroborated with our focus groups. The three major findings we elucidated were: the importance of promoting mental wellbeing through open conversations; the necessity to value frontline workers through payment of living wages, expanding employee benefits and providing PPE to protect their wellbeing; the need to leverage digital technologies to expand remote working options and access to healthcare through telemedicine/telecounseling.

Reviewer 3 Report

The authors address an interesting topic from an innovative angle, there however several shortcomings which needs to be addressed in order to make the paper more sound from an academic perspective

Abstract

It is mostly a list of recommendations and does not fully give an idea of what to be expected in the paper

Methodology

The way the qualitative and quantitative data was collected is not properly presented, it is indicated that “First …” but no indication of the second form of collection follow, probably lines 87-88 should be moved before line 82 “The Egypt/Sudan focus…” so to make clear that focus groups were the second form of data collection.

The third instrument of data collection (face to face interview) is not mention in the study design section, nor it is explained how the respondent was selected. Besides, methodologically it is difficult to understand why only one respondent from what seems to be a privileged working condition, has been chosen. Probably if the respondent was a Sudanese or Mexican informal worker who lost her job due to Covid restrictions, the outcome would have been different.

Such bias is somehow acknowledged by the authors in different part of the paper (e.g. 309-313) but little is done to effectively address it in the research. Overall, this diminishes the impact of the paper and does not really address the North South divide as in the intention of the authors.

From a technical point, since the word “interviews” is used, it should be explained if several interviews, in different periods, were conducted with the respondent.

The three major themes are not directly indicated in the section, while 4 bold catching words/themes are designated; probably it would help is fig 1 is moved above at line 149 and the 3 themes are explained. On the same breath consistency when naming in the other part of the paper the 3 themes, would provide more clarity to the reader.  

Section 3.1

The structure of the section 3.1 would benefit if lines 231-247 is moved at the beginning of the section and the respondents’ narratives are presented in such a way to make it clearer if they refer to individuals taking part to the focus group or if they are the collective analysis of the participants to the focus group. On the same line are the participants indicated in line 226 those of the Egypt/Sudan focus group (line 230) or the are they the participants of all focus groups?

The same confusion appears in other sections of the paper

Lines 268-271 probably fit better in the 3rd theme

Section 3.2

The respondents’ narratives in lines 275-289 are not well presented; as indicated above it is not clear if they refer to the focus group conclusions or are the individual narratives of the members of the groups that are reported.

Some of the statements appears imposed rather than peculiar determinants of women’s health, (e.g. provision of living wages, paid sick leave, and provision of PPE affected all frontliner workers particularly during the pandemic, regardless from the gender)

Section 3.3, 3.3.1

In section 3.3, 3.3.1. little reference is made to the overwhelming literature emphasizing the negative impact on physical and mental wellbeing of women and single mothers working from home (e.g. Yijing Xiao, Burcin Becerik-Gerber, DDes, Gale Lucas, and Shawn C. Roll, Impacts of Working From Home During COVID-19 Pandemic on Physical and Mental Well-Being of Office Workstation Users, J Occup Environ Med. 2021 Mar; 63(3): 181–190)

Section 3.3.2

The way the respondents/focus group narratives are intertwined with the authors elaboration of the same, make the section very difficult to follow

Section 4

One would have expected that recommendations were associated to the three themes emerged from the interviews/focus groups, instead they are related to 4 new categories, it would help if the authors provide a short explanation why this procedure has been followed. Most of the recommendation go beyond the gender divide

No reference in the section is provided to Fig 2 which, positioned at the end of the section appears to be decontextualized, probably it would be better find a better location and provide a short explanation of the same figure.

Section 5 does not fit very well with an academic paper, rather with a project report, in the same section it would be more appropriate to indicate Ms. Alkhadragy with her title or for example as “one of the authors ...”

Section 6.

While it is important to have acknowledged the limitation of the study, little efforts has be made to balance them. As indicated in the above comment it would be more appropriate to indicate Ms. Shroff principal author

Other general considerations

Some of the observations provided in the paper (for example lines 168, 169, “…health inequality matter less to some leaders”) do not have appropriate referencing   

The North South divide is not really taken in consideration

Throughout the paper it is indicated that the focus groups discussion confirm the deterioration of women condition during the pandemic, however such correlation with the findings is not always immediate and, in some cases, appear contradictory, for example lines 173-174 Every focus group included commentary about mental and physical health problems which have deteriorated during this pandemic” and lines 196-205.

Author Response

Reviewer 3 Comments

Author Response

Abstract

It is mostly a list of recommendations and does not fully give an idea of what to be expected in the paper

Abstract re-written to discuss more broadly our recommendations with a concluding statement that changes enacted in different parts of the world during COVID-19 have the potential to improve the lives of women in years to come.

Methodology

The way the qualitative and quantitative data was collected is not properly presented, it is indicated that “First …” but no indication of the second form of collection follow, probably lines 87-88 should be moved before line 82 “The Egypt/Sudan focus…” so to make clear that focus groups were the second form of data collection.

The third instrument of data collection (face to face interview) is not mention in the study design section, nor it is explained how the respondent was selected. Besides, methodologically it is difficult to understand why only one respondent from what seems to be a privileged working condition, has been chosen. Probably if the respondent was a Sudanese or Mexican informal worker who lost her job due to Covid restrictions, the outcome would have been different.

Such bias is somehow acknowledged by the authors in different part of the paper (e.g. 309-313) but little is done to effectively address it in the research. Overall, this diminishes the impact of the paper and does not really address the North South divide as in the intention of the authors.

From a technical point, since the word “interviews” is used, it should be explained if several interviews, in different periods, were conducted with the respondent.

The three major themes are not directly indicated in the section, while 4 bold catching words/themes are designated; probably it would help is fig 1 is moved above at line 149 and the 3 themes are explained. On the same breath consistency when naming in the other part of the paper the 3 themes, would provide more clarity to the reader. 

The methods section has been revised for clarity. Phase 1 represents study design, recruitment and survey data collection. Phase 2 represents study design, recruitment and the focus groups, the participants of which were derived from Phase 1. Phase 3 was identification and interviews of the key informant (KI). We explain in the methods section the recruitment of KI. Per the reviewers suggestions, we have clarified the language to ensure these distinct processes are clear.

We have removed the word “first”.

We have clarified that our KI was a working class person who struggled for citizenship rights, safety from violence and to raise her daughter on her own and many other challenges. She is a recipient of a temporary protection status program which makes her existence in the USA very precarious.

A key informant (KI) interview was conducted with one Latinx woman in the USA, to elucidate how the pandemic may have catalyzed better health in one of our study populations. To identify a KI from the working class Latinx community, we reached into our networks. We looked for a woman who was willing to share her story in a video interview in either Spanish or English. We found a bilingual KI who was interviewed in English to maintain consistency of the study language.

Claudia Lainez’s story offers context and insight from the perspective of a working class individual who struggles for citizenship, human rights, safety from violence and good health. These KI interviews validated the themes from the focus groups well. We felt that this one individual, interviewed twice for several hours, represented our themes exceptionally well.

Our key informant was interviewed twice. We have ethical clearance and consent from the KI for public sharing of the interviews. These are available on our website: maa.med.ubc.ca 

We have moved figure-1.

We have three themes and the last theme is broken into two sub themes. There are only three bolded themes now.

Section 3.1

The structure of the section 3.1 would benefit if lines 231-247 is moved at the beginning of the section and the respondents’ narratives are presented in such a way to make it clearer if they refer to individuals taking part to the focus group or if they are the collective analysis of the participants to the focus group. On the same line are the participants indicated in line 226 those of the Egypt/Sudan focus group (line 230) or the are they the participants of all focus groups?

The same confusion appears in other sections of the paper

Lines 268-271 probably fit better in the 3rd theme

Direct quotations are the views of individual participants. Each quotation has the name of the region from which the focus group participants came from. We have clarified these distinctions in the paper now.

We have moved many lines around to address the reviewer’s concern.

Section 3.2

The respondents’ narratives in lines 275-289 are not well presented; as indicated above it is not clear if they refer to the focus group conclusions or are the individual narratives of the members of the groups that are reported.

Some of the statements appears imposed rather than peculiar determinants of women’s health, (e.g. provision of living wages, paid sick leave, and provision of PPE affected all frontliner workers particularly during the pandemic, regardless from the gender)

We have tried to distinguish between focus group conclusions and individual quotations. 

These topics impact all genders but they definitely have a disproportionate impact on women who constitute the majority of frontline workers globally–healthcare, retail, hospitality etc.

“These recommendations will benefit many people with a disproportionate impact on those who identify as women, given that mental health burdens are greater for those who identify as women [45]. Similarly, most frontline workers in healthcare, retail, hospitality are women who have been harshly impacted by economic consequences of the pandemic [46]. Digital technologies likewise will benefit white collar women workers differently than others because most domestic and childcare labor falls to women. Finally, telemedicine and tele-counseling also benefit women significantly because they have less need to arrange for others to care for their children and loved ones if they receive services from home.”

We have added this paragraph in the Health Policy Recommendation section. 

Section 3.3, 3.3.1

In section 3.3, 3.3.1. little reference is made to the overwhelming literature emphasizing the negative impact on physical and mental wellbeing of women and single mothers working from home (e.g. Yijing Xiao, Burcin Becerik-Gerber, DDes, Gale Lucas, and Shawn C. Roll, Impacts of Working From Home During COVID-19 Pandemic on Physical and Mental Well-Being of Office Workstation Users, J Occup Environ Med. 2021 Mar; 63(3): 181–190)

We have added a sentence that acknowledges the negative impact on health status of single mothers and women and cited papers.

The sentence reads:  “Working from home during the pandemic has created many negative impacts on women’s wellbeing, overburdening them with increased childcare such as homeschooling, housework and other domestic labor in addition to their paid work; women in lower income quintiles and those who have less support experience greater levels of stress, fatigue, economic concerns, and work overload.”(Dominguez et al. 2022, Juan et Manisha Salinas 2022; Systilla 2022)”

Section 3.3.2

The way the respondents/focus group narratives are intertwined with the authors elaboration of the same, make the section very difficult to follow

We have now separated the results and discussion to make it easier to follow.

Section 4

One would have expected that recommendations were associated to the three themes emerged from the interviews/focus groups, instead they are related to 4 new categories, it would help if the authors provide a short explanation why this procedure has been followed. Most of the recommendation go beyond the gender divide

No reference in the section is provided to Fig 2 which, positioned at the end of the section appears to be decontextualized, probably it would be better find a better location and provide a short explanation of the same figure.

The 3 themes that emerged were:

  1. Improved mental health through open conversations,
  2. valuing frontline workers,
  3. digital technologies that improved the determinants' health and healthcare access.

The recommendations are

1. improve mental health through more open conversations,

2. improve conditions for frontline workers through living wages, provision of PPE, paid sick leave and benefits,

 3. Continue to provide digital technologies to support working from home and access to telemedicine and telecounseling and flow from them.

The recommendations were designed to improve women’s health. Women represent a disproportionate number of those with mental health conditions; they also predominate in front line occupations. Working from home impacts women workers who have caregiving responsibilities and housework responsibilities differently than others. Telemedicine and telecounseling is of particular benefit to women because it allows them to receive care without leaving home.

We have added the following sentences to acknowledge these:  “These recommendations emerged from our participants’ most pressing and frequently mentioned ideas combined with our contributions and evidence from scholarly literature. Figure 2 represents these ideas in a schematic form. These recommendations will  benefit many people with a disproportionate impact on those who identify as women, given that mental health burdens are greater for those who identify as women (citation). Similarly, most frontline workers identify as women (citation). Digital technologies likewise will benefit white collar women workers differently than others because most domestic and childcare labor falls to those who identify as women. Finally, telemedicine and telecounseling also benefit those who identify as women significantly because they have less need to arrange for others to care for their children and loved ones if they receive services from home. “

We have mentioned this sentence to acknowledge figure 2 “Figure 2 represents these ideas in a schematic form.”

We have provided a summary as recommended:

“As we had hoped, the focus groups generated a rich stream of dialogue and provided insight into the needs of women through the lens of expert panels on womens’ health. This was augmented by a key informant interview of a working class Latinx woman who discussed themes - challenges and needs - that corroborated with our focus groups. The three major findings we elucidated were: the importance of promoting mental wellbeing through open conversations; the necessity to value frontline workers through payment of living wages, expanding employee benefits and providing PPE to protect their wellbeing; the need to leverage digital technologies to expand remote working options and access to healthcare through telemedicine/telecounseling.“

 Section 5 does not fit very well with an academic paper, rather with a project report, in the same section it would be more appropriate to indicate Ms. Alkhadragy with her title or for example as “one of the authors ...”

We are trying to emphasize the importance of Knowledge Translation. This project was an example of policy action. 

The passage now reads: “Authors were keen to realize some concrete changes as a result of the study. 

In Egypt, one of the authors conducted sessions to help university staff cope with stress and ensure their wellbeing post-pandemic through flexible work schedules. She is also making efforts to develop accessible daycare in hospitals for frontline workers.

Maternal and Infant Health Canada, which is facilitating this study, has begun open dialog about mental health and wellbeing. In all study regions, MIHCan conducted Break the Stigma, a mental health campaign on their social media channels with the goal of destigmatizing conversations around mental health.

The section thus highlights efforts of MIHCan, the implementing body of this study, in taking initiative to implement policy recommendations outlined in the paper.

Section 6.

While it is important to have acknowledged the limitation of the study, little efforts has be made to balance them. As indicated in the above comment it would be more appropriate to indicate Ms. Shroff principal author

We have changed the sentence to “The principal author….” and taken out the name.

We have acknowledged the limitations and tried to balance them by including two in-depth interviews with a key informant who was a Latinx working class woman.  with a key informant. She articulated virtually all of the themes represented in the paper. In future research, we plan to carry out further investigations with more participants who experience poverty.

In the conclusion, there is a comment about areas for future research, which states “We plan to conduct a follow-up study on transformation and women’s health that will focus mainly on the expertise of working class women.”

Other general considerations

Some of the observations provided in the paper (for example lines 168, 169, “…health inequality matter less to some leaders”) do not have appropriate referencing  

We have added a citation for that line: American Journal of Public Health (ajph)

  1. April 2018

Mechanisms by Which Anti-Immigrant Stigma Exacerbates Racial/Ethnic Health Disparities

Brittany N. Morey PhD, MPH

We also put citations for a few other assertions.

The North South divide is not really taken in consideration

Throughout the paper it is indicated that the focus groups’ discussion confirm the deterioration of women condition during the pandemic, however such correlation with the findings is not always immediate and, in some cases, appear contradictory, for example lines 173-174 “Every focus group included commentary about mental and physical health problems which have deteriorated during this pandemic” and lines 196-205.

Our participants were experts in the field and we picked them specifically for that. We heard a great deal of congruence between participants from the Global North and the Global South.

We have put a sentence at the top of the Results Section, stating “Participants from nations in the Global North and South expressed similar ideas. This was not predicted, as socioeconomic divisions between and within nations are significant.”
